# IMPROVED IMAGE GENERATION VIA SPARSITY

## ABSTRACT

The interest of the deep learning community in image synthesis has grown massively in recent years. Nowadays, deep generative methods, and especially Generative Adversarial Networks (GANs), are leading to state-of-the-art performance, capable of synthesizing images that appear realistic. While the efforts for improving the quality of the generated images are extensive, most attempts still consider the generator part as an uncorroborated "black-box". In this paper, we aim to provide a better understanding and design of the image generation process. We interpret existing generators as implicitly relying on sparsity-inspired models. More specifically, we show that generators can be viewed as manifestations of the Convolutional Sparse Coding (CSC) and its Multi-Layered version (ML-CSC) synthesis processes. We leverage this observation by explicitly enforcing a sparsifying regularization on appropriately chosen activation layers in the generator, and demonstrate that this leads to improved image synthesis. Furthermore, we show that the same rationale and benefits apply to generators serving inverse problems, demonstrated on the Deep Image Prior (DIP) method.

## 1 INTRODUCTION

The use of Generative Adversarial Networks (GANs) for image synthesis is one of the most fascinating outcomes of the emerging deep learning era, e.g., Goodfellow et al. (2014); Radford et al. (2015); Zhu et al. (2017); Ledig et al. (2017); Karras et al. (2018). GAN is a machine-learning apparatus built of two competing neural networks, a generator and a discriminator. The generator produces samples, while its adversary, the discriminator, attempts to distinguish between them and training ones. The resolution and quality of images produced by these intriguing game-theoretic machines have improved significantly in recent years, e.g., Karras et al. (2019); Brock et al. (2019).

Although leading to impressive results, GANs are difficult to train and prone to undesired phenomena, such as mode collapse, failure to converge and vanishing gradients (Thanh-Tung & Tran, 2020). Much of the research in this field has been focusing on mitigating the above difficulties and on stabilizing the training process. Most of this progress has been achieved by heuristically modifying the architectures of the generator and the discriminator, and by exposing new and better behaved training losses (Salimans et al., 2016; Arjovsky et al., 2017; Gulrajani et al., 2017; Mao et al., 2017). As such, while GANs in general have been extensively studied and redesigned, the generator itself still operates as a "black-box" of unjustified architecture nor meaning.

Image generators within GANs are deep neural networks, comprised of a sequence of convolutional-based blocks. Distinct from other types of deep learning architectures, generators learn a transfer function from a (typically simple) multivariate source distribution to a target one. To do so, each of the generator's blocks performs a transitional step from its input to a latent representation. The desired overall mapping is thus achieved by a chain of these incremental steps. As mentioned above, previous work has considered the generator's mapping as an end-to-end process, with hardly any attempt[1] to propose a model that provides a possible explanation to the true nature of it.

Motivated by the sparse modeling literature (Elad, 2010), we aim to propose a novel interpretation that sheds some light on the architecture of image generators and provides a meaningful and effective regularization to such networks. We interpret generators as implicitly relying on sparse models in general, and on the Convolutional Sparse Coding (CSC) and its Multi-Layered (ML-CSC) version in

---

[1]A notable exception to this rule is the work reported in Mahdizadehaghdam et al. (2019), which will be described and tied to our contribution in Section 3.

particular (Szlam et al., 2010; Bristow et al., 2013; Chalasani et al., 2013; Grosse et al., 2012; Heide et al., 2015; Papyan et al., 2017b; Papyan et al., 2016; Sulam et al., 2018; Sulam et al., 2018). This observation provides a possible explanation for the generator's intermediate mappings. We harness this insight by proposing a general model-based approach to regularize image generators, which can be easily applied to various architectures.

More specifically, we refer to the generator as a machine that synthesizes a sparse representation, followed by a multiplication by a dictionary. Accordingly, we propose several versions of sparsity-insipred regularizations, posed both as constraints or penalties, injected into the GAN training process. Under the CSC interpretation, the last layer of the generator consists of the learned synthesis dictionary, built as a set of filters. ML-CSC further assumes that a sequence of such dictionaries is multiplying the representation vector for generating the output image. We validate our proposed view by conducting extensive experiments on a variety of well-known GAN architectures, from relatively simple to up-to-date ones. Experimenting with a wide range of such networks demonstrates the versatility and generality of our approach. All the conducted tests show a performance improvement (assessed via Fréchet Inception Distance and Inception Score), achieved by our simple yet effective method.

We further extend our contribution by demonstrating that the same rationale and improvement are valid for other image generator neural networks. More specifically, we apply the proposed regularizations to the Deep Image Prior (DIP) algorithm (Ulyanov et al., 2018), which utilizes an image generator as an implicit image prior for solving various inverse problems. We examine the effects of our approach, and show that in restoration tasks as well, performance improvement is attained.

Adopting a different perspective towards our contribution, image generators are highly expressive overparametrized neural networks (Balaji et al., 2021) that solve what seems to be a severely ill-posed problem. Using overparametrized models with a large capacity might lead to a poor generalization and inferior results, and especially so when trained with relatively small datasets. As such, a properly chosen regularization applied to the generator may lead to benefits in both stability and overall performance. Indeed, driven by similar reasons, regularization has been adopted for the discriminator, being overparametrized as well, e.g., Wasserstein and Spectral Normalization GANs (Arjovsky et al., 2017; Gulrajani et al., 2017; Miyato et al., 2018; Zhang et al., 2019). Our work offers to close this gap by introducing a meaningful such regularization for generators, thereby further stabilizing and boosting the synthesis process.

To summarize, this work focuses on deep neural networks that serve as image generators, and its key contributions are the following:

- We propose a novel sparse modeling interpretation of image generators for better understanding them.
- We leverage this interpretation for proposing appropriate sparsity-inspired regularizations to generators.
- We demonstrate the validity of our view by showing a performance boost in image synthesis with various GAN architectures.
- We show that a similar performance improvement is applicable to image generators serving inverse problems, in the context of DIP.

## 2  BACKGROUND

### 2.1  SPARSE MODELING

Sparse Modeling has been proven to be highly effective in signal and image processing applications, e.g., (Dabov et al., 2007; Bruckstein et al., 2009; Yang et al., 2010; Elad, 2010; Dong et al., 2011; 2013; Mairal et al., 2014). This model assumes an underlying linear generative model, according to which, a signal $x \in \mathbb{R}^N$ can be described as a linear combination of few columns from a dictionary $\mathbf{D} \in \mathbb{R}^{N \times M}$, i.e. $x = \mathbf{D}\Gamma$, where $\Gamma$ is a sparse vector. The columns of $\mathbf{D}$ are referred to as atoms and they may form an overcomplete set, i.e., $M > N$. Retrieval of a sparse vector $\Gamma$, corresponding to a given a signal $x$ and a dictionary $\mathbf{D}$, is referred to as *sparse coding*, formulated as

$$\min_\Gamma \|\Gamma\|_0 \ s.t. \ \mathbf{D}\Gamma = x, \qquad (1)$$

where $\|\Gamma\|_0$ counts the non-zeros in $\Gamma$. Thus, synthesizing a signal according to this model is done by generating a sparse representation vector $\Gamma$ and multiplying it by $\mathbf{D}$.

## 2.2 CONVOLUTIONAL SPARSE CODING MODEL

When handling images, sparse modeling has been usually deployed on small overlapping patches due to the complexity of handling a general content dictionary. The Convolutional Sparse Coding (CSC) model (Szlam et al., 2010; Grosse et al., 2012; Bristow et al., 2013; Chalasani et al., 2013; Heide et al., 2015; Papyan et al., 2017b) is a global variant of the above model, which bypasses the need to operate on patches. The CSC's dictionary is structured, being a concatenation of banded circulant matrices containing small support filters, each appearing in all possible shifts. Thus, $\mathbf{D} \in \mathbb{R}^{N \times mN}$ where $N$ is the size of the signal $x$ and $m$ is the number of filters, each of length $n \ll N$. According to this model, a signal $x$ can be expressed by $x = \mathbf{D}\Gamma$, as described above (see Figure 1).

The CSC has demonstrated superb performance in image processing tasks, such as denoising, separation, fusion, and super-resolution (Gu et al., 2015; Liu et al., 2016; Papyan et al., 2017a; Simon & Elad, 2019; Zisselman et al., 2019). A recent work (Papyan et al., 2017b) has proposed a theoretical analysis of this model, exposing the need for a redefinition of the sparsity measure to be used on $\Gamma$, in order to account for local use of atoms instead of globally counting non-zeros. We shall get back to this in the next section, when using the CSC model.

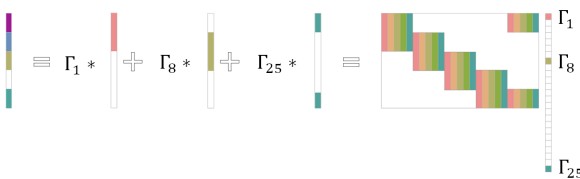

Figure 1: CSC visualization: A signal $x$ is generated by a superposition of few atoms from a convolutional dictionary $\mathbf{D}$. Each entry of $\Gamma$ corresponds to a certain shift of a limited support filter.

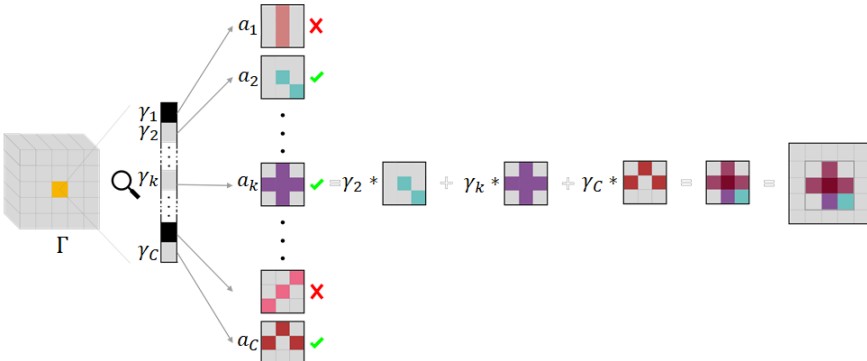

Figure 2: 2D CSC visualization: $\Gamma$ is an $H \times W \times C$ tensor, composed of $H \times W$ "needles", denoted as $\gamma$, each of size $1 \times 1 \times C$. Since $\Gamma$ is sparse, the bulk of its entries are zeros. A given $\gamma$ at location $i, j$ in $\Gamma$ contributes to a patch located in the corresponding location in the output image $x$. Each non-zero entry $\gamma_i$ in $\gamma$ defines the coefficient for the filter $a_i$ in a superposition that creates a patch. Note that the patches created by nearby needles may overlap, depending on the stride and the spatial size of the atoms in $\mathbf{D}$.

## 2.3 Multi-Layer Convolutional Sparse Coding (ML-CSC)

ML-CSC (Papyan et al., 2016; Sulam et al., 2018; Sulam et al., 2018) is an extension of the CSC, which generates a cascade of sparse representations $\{\Gamma_i\}_{i=1}^{L}$, corresponding to CSC dictionaries $\{\mathbf{D}_i\}_{i=1}^{L}$. This model assumes that a signal $x \in \mathbb{R}^N$ can be represented as

$$x = \mathbf{D}_1\Gamma_1 \;\; s.t. \;\; \|\Gamma_1\|_0 \leq \lambda_1,$$
$$\Gamma_1 = \mathbf{D}_2\Gamma_2 \;\; s.t. \;\; \|\Gamma_2\|_0 \leq \lambda_2,$$
$$\vdots$$
$$\Gamma_{L-1} = \mathbf{D}_L\Gamma_L \;\; s.t. \;\; \|\Gamma_L\|_0 \leq \lambda_L,$$

where $\{\lambda_i\}_{i=1}^{L}$ are sparsity thresholds. Thus, while the first equation perfectly aligns with the regular CSC model, the additional equations add further structure by suggesting that each sparse representation vector is by itself a CSC signal. Note that by substituting the above equations, a signal $x$ can also be described as $x = \mathbf{D}_1 \cdots \mathbf{D}_i\Gamma_i = \mathbf{D}_{eff}\Gamma$, $1 \leq i \leq L$, with intermediate sparse representations.

## 3 Improved Image Synthesis via GANs

Deep generative models are neural network-based architectures that synthesize signals, such that their output follows the probabilistic distribution of a given data source. In computer vision, these generators are used successfully for synthesizing natural images, by training on massive datasets of photos. To this end, they map a given source distribution $P_z$ (typically a simple and canonical multivariate Gaussian) to a data distribution of interest $P_x$,

$$G(z) = x_{gen} \;\; s.t. \;\; z \sim P_z \;\; and \;\; x_{gen} \sim P_x. \tag{2}$$

Due to the complexity of image synthesis, the above mapping function is usually modeled by a highly expressive feed-forward deep convolutional neural network, consisting of several consecutive layers. In its simplest and most common form, an image generator with $K$ layers can be rephrased as a feed-forward CNN of the form

$$G_K(...(G_1(z))) = x_{gen}, \tag{3}$$

where $G_i$ represents the i[th] layer of the generative model, applying convolutions, normalizations and a non-linearity (typically ReLU). Thus, the overall mapping is attained in tandem of $K$ transitional mappings. Despite the incremental nature of the deep image generator, it is treated and trained in an "end-to-end" mode, without understanding the purpose of the inner activations, nor enforcing a specific structure or properties on the intermediate activations.

In this work, we interpret existing image generators as implicitly relying on sparse modeling, and we propose a novel model-based approach to describe and improve the synthesis process of such architectures. Since these generators are highly expressive and overparametrized, confining them can lead to superior results and facilitate the training process. To do so, we recall the convolutional sparse coding (CSC) and its Multi-Layer version (ML-CSC) models, as discussed in Section 2.

According to the CSC model, an image $x$ can be represented as a multiplication of a sparse representation vector $\Gamma$ by a convolutional dictionary $\mathbf{D}$, i.e. $x = \mathbf{D}\Gamma$. The ML-CSC further assumes that $\mathbf{D}$ is achieved by a multiplication of $L$ dictionaries: $\mathbf{D}_{eff} = \mathbf{D}_1\mathbf{D}_2 \cdots \mathbf{D}_L$. Note, however, that although both are generative models, there is a substantial gap between their description and the process described in Equation (2). Whereas the CSC synthesis starts with a sparse representation vector, the image generation setup in Equation (2) starts with a dense latent random vector $z$. To bridge this gap, we propose to interpret image generators as performing two consecutive tasks:

1. $G^S$: Map the input vector $z$ to a sparse latent vector $\Gamma$ (done by the generator's first $K-1$ layers).

2. $G^I$: Multiply $\Gamma$ by a convolutional dictionary $\mathbf{D}$, i.e., $\mathbf{D}\Gamma \sim P_x$ (performed by the generator's K[th] layer that learns a convolutional dictionary for this purpose).

The second task is exactly the CSC synthesis process. More specifically, if $\mathbf{D}$ is a single convolutional dictionary, the above formulation is equivalent to the CSC, whereas if $\mathbf{D}$ is comprised of two

or more dictionaries, this becomes the ML version of it. By splitting the generation process into two parts, we identify the role of $\Gamma$ as the sparse representation in a (ML-)CSC-based model. This way, the image synthesis process can be described as

$$G^S(z) = \Gamma \ , \ G^I(\Gamma) = x$$
$$s.t. \ z \sim P_z, \ \Gamma \ is \ sparse, \ x \sim P_x,$$

namely, the end-to-end mapping can be separated into two consecutive mappings, where each may be easier to learn. This interpretation is further shown in Figure 3. In the ML-CSC context, $G^I$ is comprised of two or more global convolutional dictionaries.

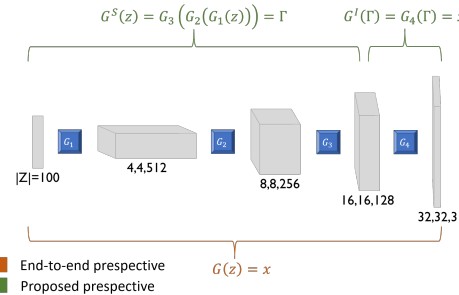

Figure 3: Our interpretation (green) of generators as handling two separate sub-tasks – $G^S$ produces a sparse representation vector and $G^I$ maps it into an image. The depicted architecture contains 4 blocks and the stated tensors' dimensions are typical for synthesizing $32 \times 32 \times 3$ images.

We have analyzed the sparsity of $\Gamma$ in regular adversarial training and found out that it is not sufficient. For the purpose of designing more compatible generators with both the CSC and the ML-CSC models, we encourage $G^S$ to map $z$ to a truly sparse representation $\Gamma$. To this end, we propose few explicit techniques, serving as regularization forces applied in the training phase:

(1) **$L_1$ regularization**: Adding an $L_1$ based penalty on the representations $\Gamma$ to the overall loss of the image generator.

(2) **$L_0$ constraint**: Eliminating small non-zero entries in $\Gamma$ to satisfy a predefined sparsity constraint: $\|\Gamma\|_0 \leq \lambda$. Namely, the amount of non-zero entries in $\Gamma$ should be less or equal to $\lambda$. This can be viewed as a projection to a constraint-satisfying tensor, obtained by zeroing the smallest absolute values of the representation.

(3) **$L_{0,\infty}$ inspired constraint**: This pseudo-norm is based on a new sparsity measure related to the CSC Papyan et al. (2017b). $\|\Gamma\|_{0,\infty}$ is the maximal number of non-zero coefficients affecting any pixel in the image $x$. Thus, forcing $\|\Gamma\|_{0,\infty} \leq \lambda$ restricts the number of local atoms to $\lambda$. While this constraint is theoretically justified in the context of CSC, projection onto it is known to be challenging (Plaut & Giryes, 2019). To approximate it in a computationally plausible manner, we use a "needle"-based sparsity measure (Papyan et al., 2017a; Zisselman et al., 2019). In the general 2D CSC case, $\Gamma$ is a 3D tensor, of size $H \times W \times C$, where H and W define the image size to be synthesized, and C is the number of filters. We define a needle as $1 \times 1 \times C$ tensor, contained in $\Gamma$. Hence, $\Gamma$ contains $H \times W$ needles, each contributes to a patch of pixels in a corresponding position in the output image. In this configuration, every pixel is affected by several adjacent needles. This setup is described in Figure 2. We propose to limit the amount of non-zero entries in every such needle. To this end, for a given representation tensor $\Gamma$, we zero the smallest absolute values in each needle of the representation to satisfy the relaxed constraint.

Although the above three options all promote sparsity in $\Gamma$, they are substantially different, as indeed will become evident in our experiments. While $L_1$ and $L_0$ consider global sparsity, the $L_{0,\infty}$ forces a local balance in the use of the atoms, i.e., limiting the local density in the representation $\Gamma$. As for the difference between $L_0$ and $L_1$, the first is deployed as a constraint, while the latter is used as a penalty. Since both $L_{0,\infty}$ and $L_0$ are constraints, we implemented them as projection layers which map an input tensor to a constraint-satisfying one, by zeroing its smallest entries. Figure 4 illustrates the effect of applying the proposed regularization techniques and the differences between them in inducing sparsity.

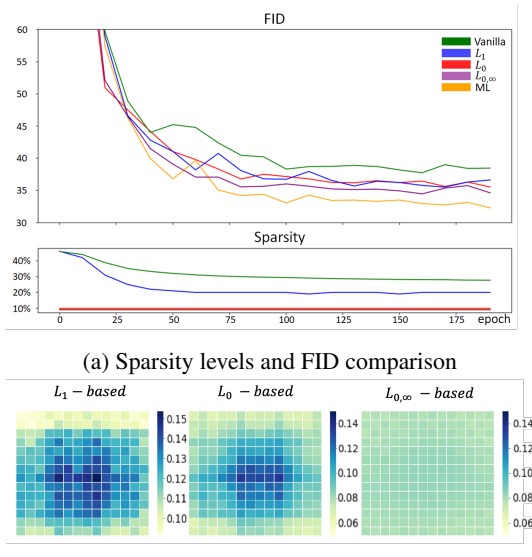

(a) Sparsity levels and FID comparison

(b) Sparsification methods' comparison

Figure 4: An illustrative experiment on the effects of applying our method during the training of a simple DCGAN (Radford et al., 2015) on CIFAR-10 dataset. (a): Sparsity levels and FID comparison between non-regularized, $L_1$-based regularization and a constraint-based regularization ($L_0$ and $L_{0,\infty}$) on a single-layer CSC and a $L_{0,\infty}$ constraint applied on the ML-CSC. The sparsity level is the percentage of non-zeros in the representation $\Gamma$. As can be seen, promoting sparsity leads to improved performance. (b): A spatial sparsity distribution comparison of $\Gamma$, obtained by the different sparsifying techniques. Each pixel in the above figure represents the mean sparsity attained in the corresponding needle of the sparse tensor $\Gamma$. As demonstrated above, $L_0$ and $L_1$ lead to a global sparsity which is imbalanced locally, while $L_{0,\infty}$ forces such a balance. Since most of the objects in CIFAR-10 are centered, applying $L_0$ or $L_1$ regularizations leads to denser needles at the center.

We should note that a connection between sparse modeling and GANs' generators has been already proposed in Mahdizadehaghdam et al. (2019). In their work, they introduced sparsity in GANs for boosting performance, and achieved their goal by designing a specific architecture that utilizes a patch-based modeling and a pre-trained dictionary. While inspired by their view, our work differs substantially, as we propose a more general strategy that leverages more advanced models (CSC and ML-CSC), does not require a pre-training of any sort, and can be easily applied to various existing GAN architectures. Indeed, as we shall see in the next section, our theme extends to other generators that go beyond GANs.

To summarize the proposed theme, we argue that learning a direct mapping from random noise to natural images' distribution is an extremely hard task that can be mitigated by enforcing a model that provides a meaningful regularization. As in every model, the CSC introduces specific structured assumptions about the data. Since this model has been shown to be highly compatible with natural images, we believe that these assumptions empirically hold, and hence, the suggested approach will lead to better results. To verify the above statement, we conduct experiments in various types of tasks, compare and dissect the results of regularized and non-regularized generators.

## 4 Improved Solution of Inverse Problems

Inverse problems involve the process of recovering data from degraded observations. There are various well-known such problems in image processing, e.g., denoising, deblurring, inpainting and super-resolution, all of which are ill-posed, as they may have no unique solution. In the field of image processing, solving inverse problems is one of the most important and studied topics, and there are thousands of papers dedicated to the derivation of various algorithms and strategies for handling such tasks. In this paper, we focus on one specific and fascinating method, DIP (Ulyanov et al., 2018), which uses an image generator for handling inverse problems.

According to DIP, a deep image generator should be trained (i.e., adapt its parameters $\theta$) to map a fixed random tensor $z$ to a given corrupted image $x_0$, and the solution to the inverse problem would be the generator's output. Formally,

$$\theta^* = \arg\min_{\theta} L(G_\theta(z)|x_0), \quad x^* = G_{\theta^*}(z), \tag{4}$$

where $L(x|x_0)$ is a task-dependant loss term. This way, much of the information about the images' distribution $P_x$ is derived from the generator's architecture.

DIP is an appealing unsupervised technique for handling inverse problems, and we find it is ideal for testing our core hypothesis – the main claim advocated in our work is that CSC based models are adequate for generating images. If indeed correct, one would expect that CNN architectures with induced sparsity could serve better as a prior for inverse problems. Recall that both $L_0$ and $L_{0,\infty}$ sparsity constraints are enforced via an architectural modification (a projection layer). DIP puts forward an exact such test, as its essence is a reliance solely on the model's architecture itself for regularizing inverse problems. Therefore, image restoration via DIP serves as a different yet meaningful setup for verifying our sparse-modeling assumptions. To this end, we aim to inject well-justified sparsity-inspired regularizations to specific activation layers within the generator for improving the final outcome.

Although DIP is a generic concept that can be applied to various generator architectures, we experiment with the same architecture as in the original paper – an encoder-decoder model that maps a latent vector drawn from $P_z$ into an image of the same spatial extent. More specifically, DIP uses a U-Net-like architecture (Ronneberger et al., 2015) with skip-connections. Due to the significant structural difference between the encoder-decoder and standard generators, applying our method in the same way in both is unjustified. To better understand the proper context of sparsity in such architectures, we focus on the encoder part of the overall system. We interpret this part as transforming its input into a set of transitional representations $\{\Gamma_1, ..., \Gamma_K\}$, where $K$ is the number of scales in the architecture, as can be seen in Figure 5. These are injected to the decoder, from which it constructs the output image. As these representations are attained by a CNN, which resembles a multi-layered thresholding algorithm (Papyan et al., 2016; Sulam et al., 2018; Sulam et al., 2018), we interpret this process as *sparse coding* serving the ML-CSC model. According to this perspective, the encoder maps the input random vector to a dense signal and proceeds by performing a multi-layer *pursuit* to obtain $\{\Gamma_1, ..., \Gamma_K\}$, in the spirit of the ML-CSC model. Using this observation, we propose to apply our sparsity-inducing regularization, as described in section 3, on all these representations.

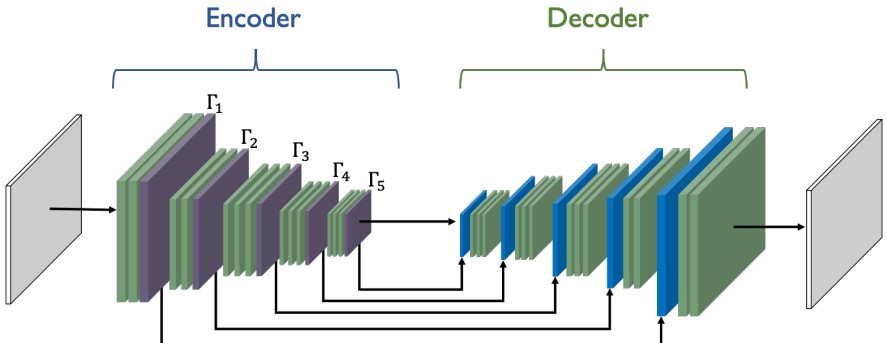

Figure 5: A visualization of the intermediate representations in an encoder-decoder architecture with skip-connections.

## 5 EXPERIMENTS

In this section we experimentally examine the proposed method, comparing its performance with non-regularized ("vanilla") image generator's architectures. First, we evaluate our method on image synthesis via GANs. We conduct an extensive study on a variety of GAN architectures and explore the effect of applying our suggested regularizations to them. In addition, we study the effect of applying our approach on limited data sources of different sizes. We also show that sparsity-inducing

regularization is versatile and can be applied to more general image generators. To this end, we apply our method to the Deep Image Prior algorithm, which handles restoration problems. In both image synthesis and in solving inverse problems, we dissect the effects of the proposed regularization techniques that induce sparsity, as described in the previous sections.

## 5.1 IMPROVED IMAGE SYNTHESIS

For experimenting with the suggested regularizations for image synthesis, we apply these on GANs during training. The proposed approach is enforced only on the generator part, as the discriminator remains untouched. We exhaustively evaluate the effect of these regularizations on various GAN variants using the CIFAR-10 dataset (Krizhevsky, 2012), since it is one of the most popular benchmarks for image synthesis.

We apply our method to several GAN architectures, conditional and unconditional, in order to show its compatibility with different synthesis processes. We experiment with a wide range of architectures, considering both the CSC and the ML-CSC variants: DCGAN (Radford et al., 2015), conditional GAN (cGAN) (Mirza & Osindero, 2014), Wasserstein GAN with Gradient Penalty (WGAN-GP) (Gulrajani et al., 2017), Mode Seeking GAN (MSGAN) (Mao et al., 2019), Spectral Normalization GAN (SNGAN) (Miyato et al., 2018), Self Attention GAN (SAGAN) (Zhang et al., 2019) and Improved Self Supervised GAN with Multiclass minimax game (SSGAN*) (Tran et al., 2019), BigGAN (Brock et al., 2019) and Differentiable Augmentation BigGAN (DiffCRBigGAN)(Zhao et al., 2020).

To evaluate the synthesis results, we use the Fréchet Inception Distance (FID) (Heusel et al., 2017), which is a widely used metric for assessing the quality of generated images. Lower FID scores indicate a better quality of the synthesized images. In addition to the FID, we also evaluate the performance of the conditional BigGAN models using Inception Score (IS) (Salimans et al., 2016), where higher values are better.

The results of the conducted experiments are summarized in Tables 1, 2. The findings attest that using our proposed regularization techniques significantly enhances the performance across all examined GAN models, from simple to more sophisticated up-to-date architectures, which demonstrate the versatility and the generality of the proposed regularizations. Furthermore, the tested ML-CSC approach using two layers (i.e. $\mathbf{D}_{eff} = \mathbf{D}_1\mathbf{D}_2$) leads to improved performance, when compared to the single-layered CSC.

Table 1: CIFAR-10 image synthesis FID results (lower is better). The ML column corresponds to GAN variants modified to fit the ML-CSC approach, as explained in the proposed method section, using the $L_{0,\infty}$ method.

| Architecture | Vanilla | $L_{0,\infty}$ | $L_0$ | $L_1$ | ML |
|---|---|---|---|---|---|
| DCGAN | 37.75 | 34.45 | 35.53 | 35.52 | **32.30** |
| cGAN | 27.64 | 26.43 | **26.06** | 26.48 | 26.41 |
| WGAN-GP | 30.06 | 28.97 | 28.51 | 30.59 | **28.04** |
| MSGAN | 24.32 | **21.72** | 23.80 | 23.91 | 21.86 |
| SNGAN | 25.50 | 25.11 | 24.81 | 24.85 | **23.63** |
| SAGAN | 18.39 | 18.23 | 18.37 | 18.48 | **18.09** |
| SSGAN* | 11.40 | 11.19 | 11.57 | 11.11 | **10.92** |
| BigGAN | 8.23 | 7.58 | **7.50** | 7.68 | 7.66 |
| DiffCRBigGAN | 6.66 | 6.22 | 6.20 | 6.31 | **5.95** |

Table 2: IS results for conditional image synthesis of CIFAR-10 using BigGAN (higher is better).

| Architecture | Vanilla | $L_{0,\infty}$ | $L_0$ | $L_1$ | ML |
|---|---|---|---|---|---|
| BigGAN | 9.28 | 9.33 | 9.33 | 9.22 | **9.31** |
| DiffCRBigGAN | 9.36 | 9.39 | **9.48** | 9.42 | **9.48** |

We turn to study the effects of our proposed regularizations when operating with limited datasets. To this end, we use 10% (5,000 images) and 20% (10,000 images) of the CIFAR-10 dataset and experiment with a BigGAN architecture. We also examine the effects of applying our method in addition to the differentiable augmentation method (Zhao et al., 2020), which provides state-of-the-art results on limited data. Table 3 demonstrates the performance improvement attained by applying our method in both of the setups.

Table 3: Image generation results (FID) on limited data sources of regularized and non-regularized BigGAN architecture.

| Dataset | BigGAN | | | | DiffCRBigGAN | | | |
|---|---|---|---|---|---|---|---|---|
| | Vanilla | $L_{0,\infty}$ | $L_0$ | ML | Vanilla | $L_{0,\infty}$ | $L_0$ | ML |
| 20% CIFAR-10 | 20.25 | 19.78 | 19.96 | **17.72** | 11.29 | 10.93 | 10.62 | **10.33** |
| 10% CIFAR-10 | 38.33 | 42.52 | **35.72** | 37.77 | 16.80 | **15.30** | 15.82 | 16.20 |

## 5.2 Improved Solution of Inverse Problems

We proceed by experimenting with our regularizations on standalone image generators, in the context of the DIP algorithm. Our goal is to compare the ability of regularized and non-regularized image generators to serve as an implicit prior for solving inverse problems.

In the conducted experiments, we solve a denoising problem with regularized and non-regularized image generators. As already mentioned in Section 4, we use an U-Net-like "hourglass" architecture with skip-connections, as in DIP (Ulyanov et al., 2018). Excluding the regularization, the compared generators have the same architecture and are being trained with exactly the same hyperparameters and setup. We conduct a similar experiment to the one in DIP, using the standard denoising dataset. [2] In this setup, an observed noisy image $x_0$ is created by adding an additive white Gaussian noise (AWGN) with standard deviation $\sigma_n$ ($\sigma_n = 25$ in our experiments). To quantitatively evaluate the denoising performance, we use the Peak signal-to-noise ratio (PSNR).

The results are reported in Table 4, where "Single" refers to the top PSNR value achieved by a single output, and "Average" is the highest PSNR value of an averaged output (obtained by an exponential sliding window over past iterations, as performed in DIP). We run the experiment multiple times to verify the statistical significance of our results.

| | **Vanilla** | $\mathbf{L_{0,\infty}}$ | $\mathbf{L_0}$ |
|---|---|---|---|
| Single | $28.74 \pm 0.03$ | $29.03 \pm 0.03$ | $\mathbf{29.22} \pm 0.06$ |
| Average | $29.88 \pm 0.02$ | $29.94 \pm 0.04$ | $\mathbf{30.21} \pm 0.05$ |

Table 4: Denoising results of regularized and non-regularized U-Net generator using DIP.

As can be seen, enforcing sparsity by our proposed methods outperform the non-regularized model. These results demonstrate that the CSC modeling assumption is valid and contributes to better regularizing the inverse problem.

## 6 Conclusions

In this work we have described simple yet effective regularization techniques for image generator architectures, which rely on sparse modeling. We demonstrate that such regularizations yield substantial improvements across a wide range of GAN architectures. In addition, we show the versatility of the approach by applying it to image generators for solving inverse problems using DIP. In this context, our regularization improves the denoising results achieved by DIP. The enhanced performance achieved by promoting sparsity in image generators, in general, testifies to the relevance of sparsity-inspired models in image synthesis. We believe that this connection can be further learned and utilized to obtain even more promising results.

---

[2]`http://www.cs.tut.fi/~foi/GCF-BM3D/index.html#ref_results`

## 7 REPRODUCIBILITY STATEMENT

In this work we have examined our proposed method both for boosting image synthesis via GANs and in improving the solution of inverse problems via DIP. In both settings, we implemented our approach using the official software in each of the papers we have referred to, and using the same hyperparameters as stated in their experiments. In addition, in all of our tests, we evaluated the performance multiple times using different seeds, so as to factor out randomness in the attained improvement. A complete code package for running all the conducted experiments and reproducing the reported results of this paper will be published upon acceptance of the paper.

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

## APPENDICES

### A. VISUALIZATION OF THE ML-CSC ATOMS

In the field of sparse coding it is a common practice to visualize the learned atoms in the CSC model in order to demonstrate their variety and richness. In Figure A, we show the atoms of the ML-CSC dictionary as obtained for DCGAN trained on the CIFAR-10 database. In order to visualize the atoms in this two-layered ML-CSC case, we need to observe both the columns of $\mathbf{D}_1$ and $\mathbf{D}_{eff} = \mathbf{D}_1 \mathbf{D}_2$ (Papyan et al., 2016; Sulam et al., 2018). In our setup, $\mathbf{D}_1$ contains 128 atoms of size $4 \times 4 \times 3$, whereas $\mathbf{D}_2$ contains 128 atoms of size $3 \times 3 \times 128$. According to the ML-CSC, each atom in $\mathbf{D}_2$ specifies how to combine atoms from $\mathbf{D}_1$ to form an atom in $\mathbf{D}_{eff}$. Therefore $\mathbf{D}_{eff}$ contains 128 atoms, where each is of size $8 \times 8 \times 3$ and created by a sparse combination of atoms from $\mathbf{D}_1$. A visualization of the dictionaries $\mathbf{D}_1$ and $\mathbf{D}_{eff}$, trained as part of a regularized ML-CSC based DCGAN on the CIFAR-10 dataset, can be viewed in the Figure A. As expected, the atoms of $\mathbf{D}_{eff}$ are more complex than the atoms of $\mathbf{D}_1$.

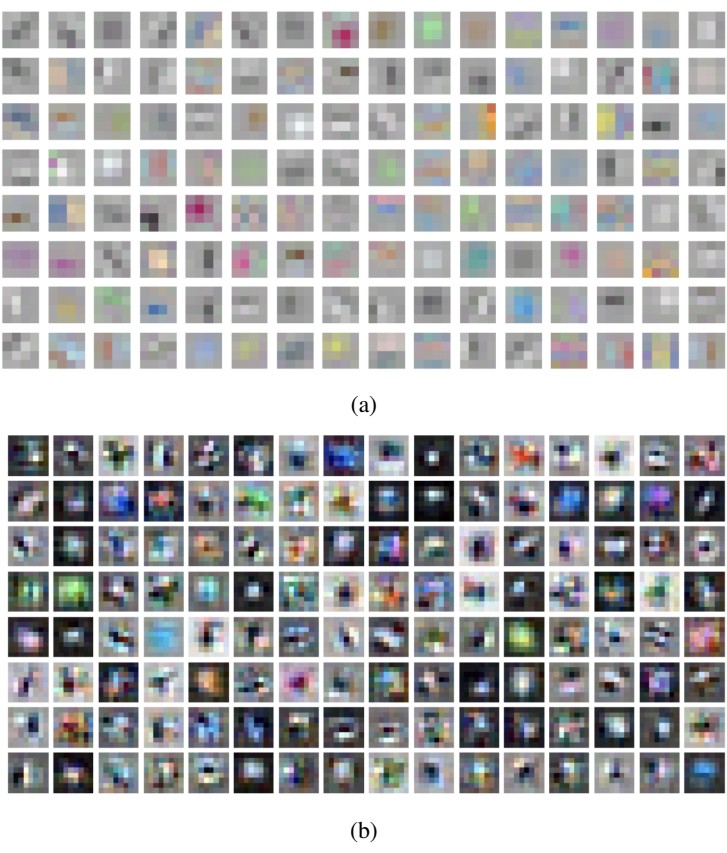

(a)

(b)

Figure A: A visualization of the dictionary atoms in the ML-CSC setup obtained for DCGAN trained on the CIFAR-10 database: (a) The 128 atoms of $\mathbf{D}_1$, each of size $4 \times 4 \times 3$. (b) The 128 atoms of $\mathbf{D}_{eff} = \mathbf{D}_1 \mathbf{D}_2$, each of size $8 \times 8 \times 3$.

### B. ADDITIONAL ANALYSIS OF SPARSITY IN GANS

We turn to present additional graphs, similar to the one given in Figure 4a, that demonstrate the effect of promoting sparsity in different GAN architectures. As can be seen in Figure B, inducing sparsity consistently improves the performance of the tested models.

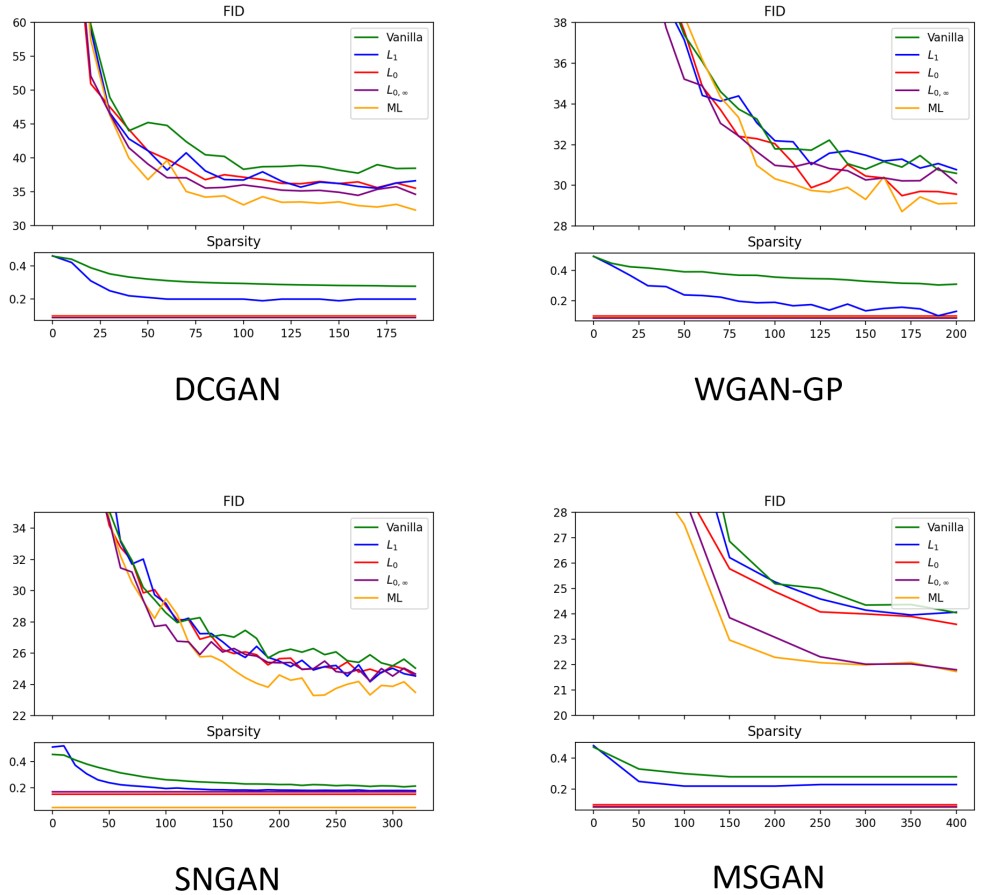

Figure B: A visualization of the performance improvement obtained by the proposed sparsity-inducing methods across various GAN architectures.

