# OpenReview forum: "Improved Image Generation via Sparsity"
_ICLR.cc/2022/Conference — ICLR 2022 Submitted_

### Official Review · Reviewer_7WVd · 2021-10-31

**Correctness:** 3
**Technical Novelty And Significance:** 3
**Empirical Novelty And Significance:** 3
**Recommendation:** 6
**Confidence:** 3

**Main Review:**

1) The paper is overall well written. I had no problem following the text, the exposition is clear and I detected minimal amount of grammar or typo issues.

2) The method is relatively simple and it could possibly be widely adopted if the experiments in the paper generalize to a wider range of problems. The paper only tested two scenarios be it with a range of baseline architectures. The paper does not discuss any downsides of the technique which I find a bit unexpected. Optimization with multiple objectives will typically introduce some kind of trade-off. In this case we are getting "better" (based on some metrics) images but at what cost? Did the sparsity change the convergence rate? Did the behavior of interpolation in the latent space change? Or is everything just strictly better?

3) The simplicity of the method could potentially be viewed as a downside of the paper since it turns up to be just a regularization scheme applied to standard architectures. E.g. Figure 5 shows standard U-Net where additional losses and/or L0 filters are applied where necessary. Since both L1 and L0 metrics are not new it makes the paper look a bit more like an experimental survey than a completely new technique.

4) I miss any qualitative comparisons of the synthesized images. The authors claim that the ML-CSC scheme leads to a significant improvement but from the tables 1,2 alone it is quite impossible to say if it is really a significant improvement or a minor improvement. Not to mention the statistical implications of the term "significant" without statistical analysis.

5) The sole reliance on PSNR as a metric without providing visual examples is even more troublesome in the DIP experiment. PSNR (as much as L2 loss) tends to be biased towards blurry results and it is generally not very well correlated with human perception of image quality. It would be fitting to use a more sophisticated image quality metric such as LPIPS or VDP. However, showing an example result would surely be the easiest option. Judging from the numbers in Table 4, I would expect the differences to be quite small, perhaps even indiscernible.

6) The paper does not provide details on how each of the baseline architecture was modified to incorporate the (ML-)CSC technique. I cannot see a supplemental document that would fill these gaps but the authors promise to provide code for all the experiments later. It is, however, not available at the time of review.


Minor editing issues:
- Abstract: "leading to state-of-the-art performance" - performance in what?
- "a meaningful such regularization" -> remove such


**Summary Of The Paper:**

The paper describes image generation as a sparse coding reconstruction process. The authors formulate the sparse coding as a core block of a convolution image generator and test several different approaches for enforcing sparsity in the trained network. The experiments show that the method can improve FID scores of common GAN networks and it can also increase PSNR of image reconstruction using deep prior when compared to the same network architectures without sparsity objective.



**Summary Of The Review:**

The paper is exploring a simple but elegant idea of formulating the typically black-box process of CNN image generation as a process of sparse dictionary decoding. I like this concept and I like that it very simply improves performance of established GAN methods. The paper is also easy to read which should make this trick accessible to the community. On the downside, the technical contribution does seem boil out to a nicely packaged sparse regularization scheme. The authors could have spent more effort analyzing the learned coding. What dictionary atoms were extracted and how meaningful they are? The experimental section covers the main check-boxes including relevant baselines and ablation of the technique but it falls short in presenting the results. The judgment of the performance is limited to a very narrow scopes of quantitative metrics and it is hard to make any conclusion about the actual magnitude of the improvement that is introduced. Furthermore, the authors also do not discuss any limitations.

Overall, after weighing all the cons and pros I am very slightly leaning towards positive outcome. It is mainly because the idea is clear, elegant and at least to the presented extent working. The evaluation is limited in scope but it appears technically valid. I would also expect the authors to address some of the concerns in the final revision.

---

> ### Author Response · Authors · 2021-11-21
> **Response**
>
> We would like to thank the reviewer for the review and constructive feedback.
>
> 1. Thanks for your comments regarding the editing issues – we will fix them in our revised version of the paper.
>
> 2. In all the conducted experiments, we ran the regularized and the non-regularized versions of the models with exactly the same hyperparameters. In the image generation setup, we trained both the regularized and the unregularized models with the same number of epochs (as used in the original papers) and evaluated the model at the end of the process. Therefore, our method does not lead to slower convergence rates.  In addition, as can be seen in Figure 4a, the trends in FID as a function on the number of epochs are similar among the regularized and non-regularized models, while the attained performance is better in the regularized ones. To summarize, according to our findings, our method leads to better performance, without any clear downsides.
>
> 3. We believe that the simplicity of our method should be viewed as a virtue since it is easy to understand and implement, and shown to be effective. Sparsity inducing regularization is not new and was widely used in previous work. However, we claim that the novelty of our work is in the way in which we apply these regularizations (in addition to the $L_{0,\infty}$ one) – a way that bridges the gap between sparse modeling and image synthesis, something that was not done before.
>
> 4. Indeed, there is no statistical analysis of each of the results in Tables 1,2, however, since we conducted many experiments over a wide range of architectures, it is clear that applying our method is beneficial. Moreover, the least significant improvement that applying our method attained was 4.2%, which is quite significant. Thank you for your comments regarding the qualitative analysis – we will add such a section to the revised version of the paper.
>
> 5. Thank you for your suggestion – we will add visual examples of the solutions for the inverse problems to the revised version of the paper.
>
> 6. Thank you for this comment. In general, for applying the CSC-based regularization, we did not modify the architecture at all, but rather added a regularization term to the loss. In the case of the $L_{0,\infty}, L_{0}$ constraint-based approaches, we added a projection layer before the last convolutional layer of the generator. The used values of $\lambda$ in these cases were chosen empirically from the range of [5, 30] - for every generative model. Empirically, the best performing values for every tested GAN were between 5% to 15% sparsity (which are significantly more sparse than non-regularized models ~30%).

---

> > ### Comment · Reviewer_7WVd · 2021-11-26
> > **Response to rebuttal**
> >
> > I thank the authors for their response.
> >
> > I believe the authors have addressed my comments within the possibilities of the media. That being said, the visual inspection of the results can only be provided in the revised paper which means it cannot be used to judge the performance of the method at the moment. Overall, the response has not lead me to change my opinion about the paper.
> >
> > Regarding the opinions of other reviewers, I agree that the simplicity of $L_0$/$L_\inf$ regularization can be viewed as both an advantage of the method and a weakness of the paper. Given that the concept is not new it clearly shifts more weight on the analytical part of the paper. There I miss some elaboration of the low-rank decomposition that is supposedly happening during the training. It is not clear whether this is really the case (as commented by the Reviewer 4xX8). The authors could analyze the learned low-rank components and whether they are meaningful, e.g. within context of dictionary representations. As it stands now, the paper just tests that the method works but does not really show why and how.

---

### Official Review · Reviewer_4xX8 · 2021-11-02

**Correctness:** 3
**Technical Novelty And Significance:** 3
**Empirical Novelty And Significance:** 3
**Recommendation:** 5
**Confidence:** 3

**Details Of Ethics Concerns:**

There  are no ethics concerns about this paper.

**Main Review:**

Strengths
1). This paper has pointed out the inefficiency of existing multi-layer convolutional sparse coding models, which is to decompose the image synthesis function into two consecutive yet independent processes. This finding is solid and backed with proper proofs.
2). Extensive experiments demonstrate that this method outperforms the state of the art baseline methods as suggested in the claims of this paper.

Weakness
1). There is one key assumption in the sparse coding that the latent code between backbone network and output is sparse when the output image is synthesised at the best performance. However, it may not be necessarily this case. The authors are suggested to prove first that the latent code layer must be a sparse vector (or  tensor)
2). Usually, PSNR or SSIM are also used for evaluating the image restoration tasks. It would be great to provide the scores of these two traditional metrics in the paper in order to have an all-around evalutions.
3). The paper does not clearly disclose the detailed parameters of the proposed method, for example, what is the selected threshold  $\lambda$ value in L0 norm? And the reason to use these threshold ? What are the weighting terms of the three different normalization constraints?

**Summary Of The Paper:**

This paper proposes a CNN-based (specifically GAN-based) solution to tackle image synthesis using sparse coding concept. The proposed method utilises the generator from a generative adversarial network to synthesise a sparse representation, i.e. the sparse code, for image synthesis. This method proposes to split original ML-CSC into two consecutive function learning. It claims that the two functions are easier to learn. In addition, this paper also proves and adds image regularisation such as DIP into the framework. Extensive experiments demonstrate that the proposed methods outperforms state of the art image synthesis methods including traditional sparse coding methods and deep learning based methods.



**Summary Of The Review:**

This paper generally proposes to improve the existing multi-layer Convolutional sparse coding frameworks by splitting the tasks into independent and consecutive components. The idea is novel and is supported by proper experiments. However, the authors are suggested to address the concerns in the weaknesses.

---

> ### Author Response · Authors · 2021-11-21
> **Response**
>
> We would like to thank the reviewer for the review and constructive feedback.
>
> 1. We propose to interpret generative models as performing a mapping into a sparse latent space and then proceeding by multiplication by CSC or ML-CSC dictionary. We empirically found out that unregularized generators do not promote enough sparsity, as can be seen in Figure 4a. We do not claim that the latent space has to be sparse, rather, due to our interpretation, we hypothesize that encouraging it to be sparse can be beneficial in terms of performance. To validate our claim, we conducted extensive experiments, both for image synthesis and image restoration, which verify that indeed, promoting more sparsity is leads to better performance.
>
> 2. Our experimental section is divided into two main parts: image synthesis and inverse problems. In the image synthesis setup, there are no ground truth images and therefore, PSNR or SSIM is irrelevant. Instead, we use FID and IS – the common evaluation metrics in the field of image generation. In the inverse problem section, the ground truth images are available and thus we evaluate our performance using PSNR, as suggested.
>
> 3. Thank you for this comment – we will add these parameters in the revised version of the paper. In general, we experiment with various values of $\lambda$ for the constraint-based approaches, from very sparse ones (5%) to ones with a similar sparsity to non-regularized ones (~30%). We performed this $\lambda$ tuning process for all GANs and found out that all of them benefit from more sparsity (the best $\lambda$ for all GANs is in range [7-15]%), a finding that verifies our initial hypothesis.

---

### Official Review · Reviewer_tsoA · 2021-11-02

**Correctness:** 2
**Technical Novelty And Significance:** 2
**Empirical Novelty And Significance:** 2
**Recommendation:** 3
**Confidence:** 3

**Main Review:**

Overall the paper is mostly well readable and understandable, and I enjoyed reading the background section on the sparse convolutional models. Perhaps I would include some more technical detail (sizes of layers and filters, regularization weight, etc) in the appendix. The FID gains also seemed potentially promising, but on closer thought I have some significant reservations about the paper, that prevent me from giving a higher score at this time.

The first half of the paper is framed around understanding and revealing the meaning of the internal activations of deep generative CNN's. As far as I can see, the findings fall short from this goal. They don't address the intermediate activations beyond the last ones, and for those, the reader doesn't really get an understanding of whether and how the meaning has changed or been elucidated. It is not enough to simply argue that sparse coding finds more meaningful features and filters, and therefore this must also happen when they are combined with CNN's. Did it actually happen and how would we tell? Can we more clearly attribute this or that meaning to different feature channels now?

The expecatation set up in the first five pages was that it would be revealed that generators have been secretly doing sparse coding all along. However this was not really the case, and it seems that the proposed "interpretation" is more like a specific way to decompose the last layers so that a sparsity prior can be inserted. No sparsity seems to be really happening by default (other than maybe via relus zeroing out values, but it's arguable if this is genuine sparsity). From there on the paper mostly focused on showing that adding the regularization improves the metrics.

Inserting the point of sparsity specifically at the end of the network seemed a bit arbitrary to me, and even if this were justified from some principle, it may be significantly limiting in practice. For a modern GAN with a reasonably high-resolution dataset (unfortunately the only experiments are on the extremely low-resolution CIFAR, more on which later), the meaning of the last layer is arguably rather minor, at least from the standpoint of sparsity of the local neighborhoods. It no longer affects any of the global structure of the image, and effectively just overlays different versions of the same image that have already been built earlier in the network. The filters themselves have very little opportunity to affect anything beyond the finest sharp detail in the full resolution. For this reason it is somewhat unclear if any "meaning" can be imposed on that end of the activations, and what exactly the role of sparsity should be there.

In the DIP experiment, as far as I can see, sparsity is applied all through the network. Why is this not something one would consider with GANs also? It's also not clear why the sparsity should apply specifically to the encoder and not the decoder -- and if it were to apply to the decoder, why wouldn't it apply to the similar looking generator also? And again, are the internal activations easier to interpret in the proposed DIP compared to the original? The metric improvement is an interesting effect, but the promised understanding of the network internals is forgotten in practice.

Figure 4 seems to indicate that there is a significant amount of sparsity even in the baseline model with no regularization. I assume this is because the relus set part of the values to zero? Given that there is such a strong source of apparent (and in my opinion, questionably meaningful) sparsity in the system already, the paper should clarify the relationship between these things. Also, what would happen with some pre-activation that didn't zero out any values, e.g., leaky relu? Wouldn't clamping the lowest ones just end up mostly cutting away the leaked tails, to little effect? What about tanh that produces both positive and negative values?

These shortcomings would be more acceptable if the results were striking in the end. It is true that some consistent improvement to the FID has been achieved in some cases, which is potentially promising and could warrant a closer examination. Unfortunately the evaluation is solely centered on the CIFAR dataset, which is a reasonable benchmark to include in a GAN paper, but hardly enough if it is the only one. Findings on such toy datasets do not always generalize to cases of actual interest. In this case, CIFAR has specific properties that may hide some issues in the proposed method. The very low resolution may work in favor of this approach, because now even the final filters have a fairly wide footprint, and they can still make meaningful contributions to the overall content of the image. But will this effect be diluted away with realistically sized images, where the filter footprints are on the order of a hundredth of the image size? (See my more detailed comments on the intermediate vs final layers, above.) Also, some very relevant methods have been excluded from the analysis, in particular the entire StyleGAN line. It would be important to know if the advantage persists there, as these are in widespread use.

**Summary Of The Paper:**

The paper proposes viewing CNN's through the lens of sparse coding. In practice it mostly boils down to encouraging sparsity in the last layer activations. To this end the paper proposes three different mechanisms that promote sparsity. This is shown to improve the FID scores somewhat in one dataset with a range of different GAN architectures. The ideas are also applied to another generator-like DIP scenario.

**Summary Of The Review:**

There are shortcomings in the overall concept as well as its evaluation. The findings suggest that this might be a promising avenue of research, but it would need to be taken further. At present, the paper boils down too much into simply adding a simple regularizer at the end and observing that it somewhat improves some metrics in a limited number of scenarios. Due to the limitations of the evaluation, it remains unclear whether the proposed improvement carries over to state of the art models and datasets. Similarly, the promised elucidation of the purpose of the feature values never really materializes.

---

> ### Author Response · Authors · 2021-11-21
> **Response**
>
> We would like to thank the reviewer for the review and constructive feedback.
>
> As correctly stated, our method does not provide a complete understanding of image generators, nor do we claim to provide such an understanding. Rather, our work provides a better understanding of generators via a sparse-modeling interpretation. More specifically, we propose a hypothesis that image generators can be viewed as relying on sparse modeling, first generating a mapping from a plain latent vector (Gaussian) into a sparse latent space, and then performing a simple linear multiplication by a CSC-based dictionary for creating the resulting image.
>
> Indeed, our proposed sparsification methods are simple, but shown to be very effective, as can be seen from our empirical experiments on a wide range of architectures (the FID values of all the architectures were improved at least by 4.2%). We believe that simplicity is the virtue of our method and not its disadvantage.
>
> We experimented only using CIFAR-10 images but used many GAN architectures to verify that our method works. Future work can indeed extend this study to higher resolution datasets, however, in this paper, we show an interesting concept and demonstrate its capabilities, both in image generation and in solving inverse problems.

---

> > ### Comment · Reviewer_tsoA · 2021-11-29
> > **Response to rebuttal**
> >
> > Thank you for the response.
> >
> > Unfortunately the rebuttal does not address most of my concerns in a significant way, and I remain on the side of rejection.
> >
> > I understand what the proposed hypothesis is, but I presented specific concerns about whether this hypothesis is plausible and whether it is supported by empirical evidence, and these were not addressed. It is not clear if the component identified as "CSC-based dictionary" is actually _behaving_ like a CSC-based dictionary in typical networks. If, on the other hand, the theoretical development is merely used to find a good spot for inserting the sparsity priors, then the claims of new understanding about generators should be removed, because the findings do not concern typical generators that don't use that regularization.
> >
> > Regarding simplicity -- it is true that avoiding unnecessary complications is good in general. However, in this case it is possible that simplicity (particularly, the extemely low resolution) is the only reason why any benefits are seen, and they could fade away in more realistic settings. I discussed why this might be so in my review, but the rebuttal did not follow up on these arguments. The effect on FID across these architectures is a potentially promising lead, but more work is needed to see if holds up.

---

> > > ### Author Response · Authors · 2021-12-02
> > > **Response**
> > >
> > > We thank the reviewer for the response.
> > >
> > > The last layer of image generators is actually a multiplication by a CSC dictionary (these are mathematically equivalent). However, the CSC and the ML-CSC require the signal to be sparse before the multiplication. We analyze existing architectures, trained without any regularization, and verify that indeed, they do not promote sparse enough signals. Therefore, we find empirical evidence that during the training of such networks, they do not learn to produce sparse signals. Thus, although performing a CSC-like operation, they do not "use the CSC correctly" since the signals are not sparse enough. We hypothesize that inducing more sparsity can improve the performance and we demonstrate this in a line of experiments.
> > >
> > > We indeed experiment with image synthesis only on low-resolution CIFAR10 images, however, we show that inducing sparsity is also beneficial to image generators on high-resolution images in image denoising using DIP.

---

### Official Review · Reviewer_ob62 · 2021-11-03

**Correctness:** 3
**Technical Novelty And Significance:** 2
**Empirical Novelty And Significance:** 2
**Recommendation:** 3
**Confidence:** 4

**Main Review:**

Strengthes:
- The proposed method is neat and did improve the performance of other GANs with the proposed structure and the loss.

Weaknesses:
- It is not easy to see clear differences between the work of Mahdizadehaghdam and the proposed work. Even though the authors argued that they are different, but there is no clear supporting arguments theoretically and experimentally. CSC / ML-CSC can be seen as an advanced version of other dictionary based methods, but it has been used in other deep learning related works (see LISTA-CPSS, ALISTA) and employing CSC / ML-CSC into the framework does not seem like a major contribution.
See also the following work:
[Zhou] Zhou et al., SPARSE-GAN: SPARSITY-CONSTRAINED GENERATIVE ADVERSARIAL NETWORK FOR ANOMALY DETECTION IN RETINAL OCT IMAGE, IEEE ISBI 2020.
- It was claimed that "We propose a novel sparse modeling interpretation of image generators for better understanding them.", but the work of Papyan et al., 2017b does not have to be limited to non-generators, so it is hard to see this as a contribution of this work.
- Results on CIFAR-10 seem too weak considering recent advanced in GANs for high resolution images. It is not easy to be convinced that the proposed method works well since images are too small (is the small image sparse?)
- DIP is not a good method for denoising and usually requires multiple realizations for ensembling to boost the performance. Moreover, DIP requires early stopping for good denoising performance, while the sparsity constraint may affect this issue instead of purely enforcing sparsity. There are also prior works on DIP using other regularizers such as the following:
[Liu] Liu et al., IMAGE RESTORATION USING TOTAL VARIATION REGULARIZED DEEP IMAGE PRIOR, IEEE ICASSP 2019.
[Mataev] Mataev et al., DeepRED: Deep Image Prior Powered by RED, ICCVW 2019
So, for fair comparisons, it should be compared with these instead of vanilla DIP.

Comments:
- How is the sparsity guaranteed during testing?
- It is recommended to show images for visual comparisons.

**Summary Of The Paper:**

This work proposes a sparsity enforcing structure for GANs by splitting layers into the generator for the sparse vector and for the final image and by training with sparsity enforcing regularizer in the latent space between them. The proposed method yielded improved performance over conventional methods in FID, IS for generators and PSNR for denoising with DIP.

**Summary Of The Review:**

The idea of this work is to enforce sparsity in CSC domain in the latent space of GANs and this work showed the effectiveness of it. However, this idea is very close to the work of Mahdizadehaghdam and there are other works to promote sparsity in latent space such as [Zhou]. Moreover, it was tested with very small images, so it is not easy to see this work working well and the experiment with DIP seems to contain some potential issues in terms of iteration number and fairness.

---

> ### Author Response · Authors · 2021-11-21
> **Response**
>
> We thank the reviewer for his comments.
>
> The work of Mahdizadehaghdam is indeed related to ours and has motivated us. However, there are some important differences that we would like to highlight: They use a specific architecture that utilized patch-based sparsity and required a pretrained dictionary. In contrast, our work is more generic and can be applied to any GAN architecture, it uses more advanced sparse models (CSC and ML-CSC), and performs the entire dictionary learning process during the optimization of the system (thus, our method does not rely on any pretrained dictionary).
>
> As mentioned correctly, CSC and ML-CSC are sparse models that are commonly used in previous work, such as ALISTA and LISTA-CPSS. However, we emphasize that in all these works, sparse modeling did not serve for image generation. In fact, as we claim in our work, there is a significant gap between the deployment of the CSC model and image generation: former papers require access to an input image (and there is no input image in image generation settings). Due to similar reasons, [2] is indeed limited to non-generators since it assumes that the CNNs perform sparse coding on the input image (and again, there is no input image for generators).
>
> Inducing sparsity in deep learning is indeed very common. [1] uses sparsity regularization on latent features of a specific architecture for better anomaly detection. Their sparsity has nothing to do with CSC and ML-CSC and it is not motivated by the sparse modeling theory. In contrast, our work proposes a general method for improving the performance of a completely different application – image synthesis, while being motivated by sparse modeling theory. In addition, our method provides an interpretation of general GAN architectures that [1] simply does not have.
>
> To summarize, various previous works use CSC and ML-CSC, and few use sparsity in GANs, however, we are the first to combine the two and to show that this connection leads to improved performance.
>
> Regarding our usage of the DIP algorithm, we use it as the ideal testing platform for our hypothesis – regularized generators act as better priors than non-regularized ones. Thus, the goal of using DIP is not to attain SOTA performance in solving inverse problems, but rather to verify if our claim regarding CSC and ML-CSC-based regularization is correct. Regarding the concerns regarding fairness and the number of iterations, we ran DIP and our regularized DIP exactly in the same way with the same realizations. As stated correctly, DIP uses early stopping and the sparse regularization may affect it and potentially impair fairness. To avoid this and to ensure a fair comparison, we take the best denoising results for each method (instead of applying early stopping), which truly reflect the capabilities of each method.
>
> We experimented only using CIFAR-10 images but used many GAN architectures to verify that our method works. Future work can indeed extend this study to higher resolution datasets, however, in this paper, we show an interesting concept and demonstrate its capabilities, both in image generation and in solving inverse problems.
>
>
> [1] Zhou et al., SPARSE-GAN: SPARSITY-CONSTRAINED GENERATIVE ADVERSARIAL NETWORK FOR ANOMALY DETECTION IN RETINAL OCT IMAGE, IEEE ISBI 2020.
>
> [2] Papyan et. al. Convolutional Neural Networks Analyzed via Convolutional Sparse Coding, JMLR 2017.

---

> > ### Comment · Reviewer_ob62 · 2021-11-30
> > **I would like to thank the authors for their responses.**
> >
> > I would like to thank the authors for their responses. Now I see the differences between the proposed method and other previous works better. However, it is still not clear why these differences have advantages over other choices. They should be clearly demonstrated theoretically or experimentally.
> >
> > It seems that the authors and this reviewer may have different views on image generators. In my view, the input random vector to the image generator does not have to be a purely random, but it could be replaced with other vectors such as degraded image (e.g., noisy image for clean image). Thus, it may not be easy to see the claimed contribution as an important contribution due to many other works that were mentioned in my review.
> >
> > It may be true that most other image generator works used CIFAR-10. However, there are GAN works that used much higher resolution  images - for example, BigGAN used ImageNet datasets with 128x128 - 512x512, high resolution images. Moreover, the key application of the image generator in this manuscript is denoising - I don't think it is common to use CIFAR-10 for denoising. This could be serious since its small images may not be sparse. The authors claimed "our method provides an interpretation of general GAN architectures that [1] simply does not have," but due to small images, it is not easy to see this aspect in the learned atoms in the manuscript unlike other CSC based works. This aspect is the most critical.
> >
> > After the rebuttal, I am considering to increase my score +1, but I can't still support its publication due to limited experiments that may not be able to be seen as sparse images. Using original high resolution datasets for various GANs is important to demonstrate the effectiveness of the proposed method over conventional works. Showing why the proposed method is unique is also important - for example, the proposed method can train CSC atoms during training. Then, what are the advantages over using pre-trained dictionaries?

---

> > > ### Author Response · Authors · 2021-12-02
> > > **Response**
> > >
> > > Thank you for your comments.
> > >
> > > * We are the first to use CSC and ML-CSC models for image synthesis since there is a substantial gap between the image generation setup and the sparse-modeling setup. To bridge this gap, we propose a novel interpretation of image generators. Based on our proposed view, we analyze existing generators and find out that they do not attain enough sparsity. Thus, we hypothesize that promoting more sparsity can improve the performance, and we demonstrate it empirically, both on image generation and on solving inverse problems.
> > >
> > > * In our work, we focus on image synthesis - a setup in which the input of the image generator is random. In our view, if the given input is for example degraded image, the task is solving an inverse problem and not pure generation. In image synthesis setup, we differ from the aforementioned works substantially, as explained in our previous response.
> > >
> > > * We experiment both on image generation and image denoising. In image generation, we indeed, do not experiment with ImageNet and focus on CIFAR-10. However, we conduct extensive experiments over a wide range of architecture, to verify that our method is indeed beneficial. In all of the tested architectures, our technique leads to a significant improvement. In image denoising, we experiment on a standard denoising dataset (https://webpages.tuni.fi/foi/GCF-BM3D/index.html#ref_results) and not CIFAR-10. This dataset contains high-resolution images.

---

### Decision · Program_Chairs · 2022-01-20

**Decision:**

Reject

**Comment:**

This paper introduces sparse modeling-inspired regularizations to improve deep neural network-based image generators. Experimental results on both (low-resolution) image synthesis and deep image prior-based inverse problems are used to validate the proposed method.
The majority of the reviewers were against the acceptance of the paper. As summarized by Reviewer tsoA: "There are shortcomings in the overall concept as well as its evaluation. The findings suggest that this might be a promising avenue of research, but it would need to be taken further. At present, the paper boils down too much into simply adding a simple regularizer at the end and observing that it somewhat improves some metrics in a limited number of scenarios. Due to the limitations of the evaluation, it remains unclear whether the proposed improvement carries over to state of the art models and datasets. Similarly, the promised elucidation of the purpose of the feature values never really materializes." The AC agrees with that summarization and recommends rejection.